# Pharmacological Nature of the Purinergic P2Y Receptor Subtypes That Participate in the Blood Pressure Changes Produced by ADPβS in Rats

**DOI:** 10.3390/ph16121683

**Published:** 2023-12-03

**Authors:** Roberto C. Silva-Velasco, Belinda Villanueva-Castillo, Kristian A. Haanes, Antoinette MaassenVanDenBrink, Carlos M. Villalón

**Affiliations:** 1Departamento de Farmacobiología, Cinvestav-Coapa, Czda. de los Tenorios 235, Col. Granjas-Coapa, Deleg. Tlalpan, Ciudad de México 14330, Mexico; roberto.silva@cinvestav.mx (R.C.S.-V.); belyndavc@hotmail.com (B.V.-C.); 2Department of Clinical Experimental Research, Glostrup Research Institute, Copenhagen University Hospital—Rigshospitalet, Nordstjernevej 42, 2600 Glostrup, Denmark; kristian.agmund.haanes@regionh.dk; 3Department of Biology, Section of Cell Biology and Physiology, University of Copenhagen, Universtitetsparken 13, 2100 Copenhagen Ø, Denmark; 4Division of Vascular Medicine and Pharmacology, Erasmus MC University Medical Center Rotterdam, P.O. Box 2040, 3000 CA Rotterdam, The Netherlands; a.vanharen-maassenvandenbrink@erasmusmc.nl

**Keywords:** ADPβS, blood pressure, pithed rat, purinergic receptors, vasodepressor responses, vasopressor responses

## Abstract

Purine nucleosides (adenosine) and nucleotides such as adenosine mono/di/triphosphate (AMP/ADP/ATP) may produce complex cardiovascular responses. For example, adenosine-5′-(β-thio)-diphosphate (ADPβS; a stable synthetic analogue of ADP) can induce vasodilatation/vasodepressor responses by endothelium-dependent and independent mechanisms involving purinergic P2Y receptors; however, the specific subtypes participating in these responses remain unknown. Therefore, this study investigated the receptor subtypes mediating the blood pressure changes induced by intravenous bolus of ADPβS in male Wistar rats in the absence and presence of central mechanisms with the antagonists MRS2500 (P2Y_1_), PSB0739 (P2Y_12_), and MRS2211 (P2Y_13_). For this purpose, 120 rats were divided into 60 anaesthetised rats and 60 pithed rats, and further subdivided into four groups (*n* = 30 each), namely: (a) anaesthetised rats, (b) anaesthetised rats with bilateral vagotomy, (c) pithed rats, and (d) pithed rats continuously infused (intravenously) with methoxamine (an α_1_-adrenergic agonist that restores systemic vascular tone). We observed, in all four groups, that the immediate decreases in diastolic blood pressure produced by ADPβS were exclusively mediated by peripheral activation of P2Y_1_ receptors. Nevertheless, the subsequent increases in systolic blood pressure elicited by ADPβS in pithed rats infused with methoxamine probably involved peripheral activation of P2Y_1_, P2Y_12_, and P2Y_13_ receptors.

## 1. Introduction

The cardiovascular system has the capacity, by different mechanisms, to regulate the homeostatic conditions that maintain hemodynamic variables, such as blood pressure and heart rate, within normal values [1,2,3]. As previously established [4,5,6], blood pressure: (i) directly depends on cardiac output and peripheral vascular resistance; and (ii) ensures, through the systemic circulation, a suitable distribution of nutrients and oxygen to all organs/tissues/cells in the body. Given its physiological and pathophysiological importance, blood pressure is strictly controlled through the integration of: (i) central nervous system (CNS) mechanisms, which modulate the autonomic and sensory outputs to the heart and resistance blood vessels [4,5,6,7]; and (ii) peripheral mechanisms at the level of the neuro-effector junction, including but not limited to perivascular sympathetic and sensory nerves, circulating hormones, and locally generated mediators (e.g., endothelium-derived factors, vasoactive metabolites, autacoids, etc.) [4,5,6,7].

Interestingly, perivascular sympathetic and sensory nerves can release, besides noradrenaline (producing vasoconstriction [4,5]) and calcitonin gene-related peptide (CGRP; producing vasodilatation [6,8]), respectively, several cotransmitters (e.g., nucleosides and nucleotides) that elicit cardiovascular responses per se and/or modulate the release of these neurotransmitters [9,10,11]. For instance, adenyl nucleotides including adenosine 5′-triphosphate (ATP) can be released from: (i) perivascular sympathetic and sensory nerves as a cotransmitter and, relying on several factors including the experimental conditions, can produce vasoconstrictor/vasodilator responses [4,6,7,8,10]; and (ii) arteries, veins, and the heart, as well as other organs/tissues/cells (including endothelium, platelets, erythrocytes, etc.) upon activation [6,7,10,12]. Once released, ATP is rapidly hydrolysed/dephosphorylated by several families of ectonucleotidases to adenosine 5′-diphosphate (ADP), adenosine 5′-monophosphate (AMP), and adenosine [13]. These dephosphorylated metabolites, and ATP itself, were reported in 1950 to produce diverse cardiovascular effects (including coronary vasodilatation) in several species [14].

To complement the aforementioned cardiovascular findings, in 1961 Gordon et al. [15] reported that intravenous (i.v.) administration of adenosine, AMP, ADP, and ATP in anaesthetised rats produced dose-dependent vasodepressor responses, with a range of potency where ADP was, equimolarly: (i) ~147 times more potent than adenosine; and (ii) ~45 times more potent than AMP or ATP. These authors also emphasized that the rat (compared to other larger mammals) is more sensitive to the vasodepressor action of these compounds, probably due to its much shorter pulmonary circulation time [15]. Likewise, in 1978 Dejana et al. [16] investigated the contribution of platelets to ADP-induced vasodepressor responses in anaesthetised rats and showed that: (i) rapid i.v. injections of ADP (1–30 µg/kg) induced dose-dependent and aggregation-independent vasodepressor responses; and (ii) when slow infusions of higher doses of ADP (6 mg/kg i.v., for 10 min) were given, platelet aggregation appeared to play a greater role. However, these studies [15,16] did not explore the nature of the receptors involved.

In fact, thus far, very few studies have analysed the purinergic receptors involved in ADP-induced blood pressure changes. Only a recent publication [17] has preliminarily examined the effects of adenosine-5′-(β-thio)-diphosphate (ADPβS; a stable and non-hydrolysable analogue of ADP) on blood pressure in anaesthetised rats with undamaged vagus nerves. Specifically, a single i.v. bolus of a supramaximal dose of ADPβS (330 µg/kg) elicited (in order of appearance): (i) an immediate, pronounced, short-lived vasodepressor response; and (ii) a late, weak, and long-lasting vasopressor response [17]. Significantly, only the late vasopressor response was blocked by MRS 2211, a P2Y_13_ receptor antagonist, but the initial vasodepressor response remained unchanged [17]; however, an in-depth pharmacological analysis of the purinergic P2Y receptors that participate in these blood pressure responses by ADPβS under different experimental conditions was not performed in this study.

Pharmacologically speaking, ADPβS is a preferential agonist at purinergic P2Y_1_, P2Y_12_, and P2Y_13_ receptors [11,18,19,20,21], and these receptors are highly expressed in the cardiovascular system [10]. In this context, some lines of evidence suggest that the purinergic P2Y_1_ receptor plays a role in blood pressure regulation, namely: (i) the blockade of this receptor in neurons leads to decreased peripheral chemoreceptor-mediated activation, impacting blood pressure control [22]; and (ii) this receptor participates in urinary NaCl excretion under high-sodium diets [23] and decreases pulmonary arterial pressure in pulmonary hypertension cases [24]. However, the direct effects resulting from activation of peripheral P2Y_1_ receptors have not been investigated in detail.

Remarkably: (i) P2Y_1_ receptor deficiency in diabetic rat models, particularly in mesenteric arteries, suggests its potential role in diabetes-related vascular complications [25]; and (ii) P2Y_1_ receptor knockout mice show reduced atherosclerosis markers, emphasizing its importance in vascular health [26]. Based on the above findings, this study has delved deeply into: (i) the dose–response relationship with ADPβS, using a wide range of i.v. doses to produce dose-dependent changes in blood pressure; (ii) the possible role of central (in anaesthetised rats) and/or peripheral (in pithed rats) mechanisms that participate in the blood pressure responses to i.v. ADPβS; and (iii) the pharmacological properties of the receptors that play a role in these blood pressure responses by means of the use of several doses of the P2Y receptor antagonists MRS2500 (P2Y_1_), PSB0739 (P2Y_12_), and MRS2211 (P2Y_13_) [9,10,11,18,27].

## 2. Results

### 2.1. Haemodynamic Parameters under Baseline Conditions

For greater clarity, it is worth highlighting in this section that: (i) diastolic blood pressure (DBP; mm Hg) is a more precise measure of peripheral vascular resistance and systemic vascular tone as it reflects the pressure in the arteries when the heart is resting in between heartbeats [1,28,29]); and (ii) systolic blood pressure (mm Hg) is a non-invasive index of cardiac (left ventricle) contractility as it represents the pressure in the arteries when the heart beats (i.e., when the left ventricle contracts) [1,28,29]. Accordingly, the values of systolic blood pressure, DBP, and heart rate under baseline conditions were assessed in the 120 rats, which were subdivided into four groups (see Section 4.2 General methods). These baseline values were: (i) Group 1 (i.e., anaesthetised rats without vagotomy: 117 ± 5 mm Hg, 105 ± 5 mm Hg and 345 ± 5 beats/min; *n* = 30); (ii) Group 2 (i.e., anaesthetised rats with bilateral vagotomy: 118 ± 3 mm Hg, 103 ± 2 mm Hg and 352 ± 4 beats/min; *n* = 30); (iii) Group 3 (i.e., pithed rats: 49 ± 4 mm Hg, 33 ± 2 mm Hg and 296 ± 3 beats/min; *n* = 30); and (iv) Group 4 (i.e., pithed rats continuously infused [i.v.] with methoxamine [an α1-adrenergic agonist]: 125 ± 4 mm Hg, 115 ± 4 mm Hg and 303 ± 4 beats/min; *n* = 30).

However, considering the vasodilator/vasodepressor effects of ADPβS described above (see Introduction), it is worthy of note that: (i) the changes in DBP and heart rate induced by ADPβS were determined in all animals before and after the treatments; and (ii) the changes in systolic blood pressure induced by ADPβS were additionally determined only in pithed rats with a methoxamine infusion before and after the different treatments, as only in this group did ADPβS produce an increase in this parameter following the decrease in DBP (see below).

### 2.2. Immediate Effects of Vehicle or ADPβS on Diastolic Blood Pressure

Figure 1 illustrates the initial responses elicited by consecutive i.v. bolus injections of either a vehicle (bidistilled water, 1 mL/kg given seven times) or ADPβS (0.3, 0.56, 1, 1.8, 3, 5.6, and 10 µg/kg), i.e., dose–response (D–R) curves, on blood pressure in: (i) anaesthetised rats without bilateral vagotomy (Group 1); (ii) anaesthetised rats with bilateral vagotomy (Group 2); (iii) pithed rats (Group 3); and (iv) pithed rats continuously infused (i.v.) with 20 µg/kg.min of methoxamine (Group 4). Unlike the i.v. bolus of the vehicle, which produced insignificant effects on blood pressure (if any), the i.v. bolus of ADPβS induced dose-dependent decreases in DBP (i.e., vasodepressor responses) in the four groups. Interestingly, in Group 4, where the low values of blood pressure in pithed rats was restored by a methoxamine infusion, the primary decreases in DBP by ADPβS were followed by predominant increases in systolic blood pressure with insignificant effects on DBP. These secondary vasopressor responses were not dose-dependent and appeared to show a tachyphylactic (fading) pattern. This experimental condition with a restored systemic perivascular tone helps explain why the ADPβS responses were quantitatively smaller in Group 3 (pithed rats without a methoxamine infusion), since baseline blood pressure values were lower (see baseline values above in Section 2.1 and in Figure 1).

### 2.3. Diastolic Blood Pressure Values 10 min after Vehicle, ADPβS, or Antagonists

Table 1 illustrates the values of DBP in the four groups of rats 10 min after i.v. treatment with vehicle (bidistilled water), ADPβS, or antagonists. Clearly, none of these treatments produced a significant effect (*p* > 0.05) on DBP compared with that of: (i) the control animals (*p* > 0.05); or (ii) its respective baseline value (before any treatment; not presented for greater clarity), but the baseline values in each group are shown above in Section 2.1). It is to be noted that the baseline values of heart rate have previously been reported to remain without significant effects (*p* > 0.05) after administration of these compounds in pithed rats [21,30]. For simplicity, we decided not to show these heart rate data.

### 2.4. The Immediate Effects of Vehicle or ADPβS on Diastolic Blood Pressure

Figure 2 depicts that the consecutive i.v. bolus of ADPβS (0.3, 0.56, 1, 1.8, 3, 5.6, and 10 µg/kg) elicited dose dependent decreases in DBP (D–R curves) in the four groups (*n* = 6 for each group). These vasodepressor responses were significant (*p* < 0.05) compared to the null effects of the vehicle (bidistilled water, 1 mL/kg given seven times). Notably, these responses in all four groups were consistent and remained clearly unaffected (*p* > 0.05) when the D–R curves were repeated two more times (Figure 2). This finding allows us to analyse the pharmacological properties of these ADPβS responses by using two doses of a given antagonist.

### 2.5. Effect of the P2Y_1_ Receptor Antagonist, MRS2500, on the Vasodepressor Responses to ADPβS

Figure 3 illustrates the vasodepressor responses produced by i.v. bolus injections of ADPβS before (control responses) and after i.v. treatment with MRS2500 (100 and 300 µg/kg; a P2Y_1_ receptor antagonist) in the four groups of rats. Hence, the vasodepressor responses to ADPβS were: (i) markedly blocked by 100 µg/kg MRS2500 and abolished by 300 µg/kg MRS2500 in Group 1; and (ii) practically abolished by 100 µg/kg MRS2500, with 300 µg/kg MRS2500 producing no further effects in Groups 2, 3 and 4. It is to be noted that, in Group 3, a small vasodepressor response (non-dose dependent) was maintained after 300 µg/kg MRS2500.

### 2.6. Effect of PSB0739 (a P2Y_12_ Receptor Antagonist) or MRS2211 (a P2Y_13_ Receptor Antagonist) on the Vasodepressor Responses to ADPβS

Continuing with our pharmacological analysis, Figure 4 and Figure 5 depict, in the four groups, the vasodepressor responses elicited by i.v. bolus of ADPβS before (control responses) and after i.v. treatment with: (i) PSB0739 (100 and 300 µg/kg; Figure 4); and (ii) MRS2211 (1000 and 3000 µg/kg; Figure 5). Clearly, the vasodepressor responses to ADPβS were not significantly affected (*p* > 0.05) after PSB0739 (Figure 4) or MRS2211 (Figure 5).

### 2.7. The Secondary Systolic Vasopressor Responses to ADPβS (Increases in Systolic Blood Pressure) in Pithed Rats Continuously Infused (i.v.) with Methoxamine (Group 4)

As previously shown in Figure 1, only in Group 4 were the vasodepressor responses to ADPβS followed by longer-lasting secondary vasopressor responses. Figure 6 shows that these ADPβS-induced vasopressor responses: (i) consisted of predominant increases in systolic blood pressure (left panel), with insignificant effects on DBP (right panel); (ii) were not dose-dependent and appeared to show a tachyphylactic (fading) pattern (left panel); and (iii) were reproducible when repeating the vasopressor D–R curve to ADPβS for the second time (*p* > 0.05 versus the first [control] D–R curve), but were slightly (though significantly; *p* < 0.05) attenuated when repeating this D–R curve for the third time, contrasted with the first (control) D–R curve. Moreover, the i.v. injections of the vehicle did not elicit effects on systolic or diastolic blood pressure (Figure 6).

### 2.8. The Effects of the P2Y Receptor Antagonists MRS2500 (P2Y_1_), PSB0739 (P2Y12), or MRS2211 (P2Y_13_) on the Systolic Vasopressor Responses to ADPβS in Group 4

Figure 7 shows that after i.v. administration of the above antagonists, the increases in systolic blood pressure elicited by the above i.v. bolus of ADPβS in Group 4 were: (i) practically abolished after MRS2500 (100 and 300 µg/kg; left panel); and (ii) dose-dependently blocked, but not abolished, after PSB0739 (100 and 300 µg/kg; middle panel) or MRS2211 (1000 and 3000 µg/kg; right panel).

## 3. Discussion

### 3.1. General

ADP is a diphosphate nucleotide that plays an important role in cardiovascular modulation mainly by interacting with purinergic P2Y receptors [31,32,33]. Its non-hydrolysable analogue, ADPβS: (i) preferentially stimulates P2Y_1_, P2Y_12_, and P2Y_13_ receptors [11,20] with affinity values of, respectively, 5.6, 7.5, and 7.5 [18,21]; and (ii) can elicit, when given at a supramaximal i.v. dose (330 µg/kg) in anaesthetised rats with undamaged vagus nerves, a biphasic blood pressure response. This biphasic response comprises a pronounced, transient vasodepressor response (which was not further investigated) followed by a weak, long-lasting vasopressor response (mediated by MRS2211-sensitive P2Y_13_ receptors) [17]. In an attempt to gain in-depth insight into the pharmacological characteristics of the receptors that participate in these responses (with selective antagonists at P2Y_1_, P2Y_12_, and P2Y_13_ receptors), in the present study a wide range of ADPβS doses (i.e., 0.3, 0.56, 1, 1.8, 3.1, 5.6, and 10 µg/kg, i.v.) was used (see Section 4.3 of methods), in: (i) anaesthetised rats without (Group 1) and with bilateral vagotomy (Group 2) to explore the potential role of vagal and other baroreflex compensatory mechanisms of central origin, triggered by blood pressure changes [17,34,35,36], as the CNS is intact; and (ii) pithed rats without (Group 3) and with a methoxamine infusion (Group 4) for systemic vascular tone restoration to confirm peripheral mechanisms, as the CNS is not functional [30,37,38].

### 3.2. Diastolic Blood Pressure Values Remained Unaffected after Administration of Vehicle, ADPβS, or Antagonists

The fact that DBP values in the four groups remained unaffected (*p* > 0.05) 10 min after the vehicle, ADPβS, or antagonists (compared with those of control animals or their respective baseline values; see Section 2.1 and Table 1) indicates that the immediate effects of ADPβS on blood pressure are attributable to its direct actions on (cardio) vascular P2Y receptors, as reported in 2019 by Haanes et al. [17].

### 3.3. Profile of the Responses Produced by ADPβS in the Four Groups of Animals

Since ADPβS produced dose-dependent decreases in DBP in all four groups of animals regardless of their response profile (Figure 1), we can suggest that these responses do not rely on the presence (anaesthetised rats) or absence (pithed rats) of active central mechanisms. Accordingly, this stable analogue elicited dose-dependent decreases in peripheral vascular resistance and, consequently, vasodilator responses at the systemic level [1,28,29]. As previously stated, Groups 1 and 2 (anaesthetised rats) have a functional CNS with active baroreflex compensatory mechanisms, while Groups 3 and 4 (pithed rats) do not; hence, we can hypothesise, in simple terms, that the potency of ADPβS to produce vasodepressor responses in anaesthetised rats is so high that it may even overshadow any baroreflex compensatory mechanisms of central origin triggered by the decreases in DBP [1,3,4,5,6].

Interestingly, in Group 4, where peripheral vascular resistance was restored at the postjunctional level by using pithed rats continuously infused with the α_1_-adrenergic agonist methoxamine [30,37,38], the profile of the ADPβS responses was different; namely, the vasodepressor responses to ADPβS were followed by secondary vasopressor responses consisting of predominant increases in systolic blood pressure. This latter finding: (i) leads us to infer that ADPβS may also increase cardiac contractility, as systolic blood pressure is considered an index cardiac (left ventricle) contractility [1,28,29]; and (ii) may have triggered baroreflex compensatory mechanisms of central origin in anaesthetised rats, which could have counteracted the increases in systolic blood pressure [1,3,4,5,6]. This line of reasoning may help explain why these increases in systolic blood pressure: (i) were not observed in anaesthetised rats (groups 1 and 2); and (ii) were unmasked in Group 4 (Figure 1). Significantly, the fact that these increases in systolic blood pressure by ADPβS were not dose-dependent and showed a tachyphylactic (fading) pattern is probably related to desensitization mechanisms of cardiac (left ventricle) P2Y receptors.

Furthermore, it is noteworthy that baseline values of blood pressure in Group 3 (pithed rats) are at such a low level (see Section 2.1) that ADPβS can only produce small, but still dose-dependent, vasodepressor responses in the original experimental tracings (Figure 1). However, these vasodepressor responses are much more noticeable when calculated as % change in DBP (Figure 2).

### 3.4. Reproducibility of the Vasodepressor Responses Elicited by ADPβS

The fact that the dose-dependent vasodepressor responses caused by ADPβS, but not by the vehicle, were immediate in the four groups of rats (Figure 2) suggests the possible role of peripheral (mainly endothelial) mechanisms [20] rather than direct central mechanisms. This suggestion is consistent with: (i) our results demonstrating that the vasodepressor responses to ADPβS were also elicited in pithed rats (without or with a methoxamine infusion; Figure 2), in which the CNS is not operative [30,37,38]; and (ii) the hydrosolubility of ADPβS (and its stability, so it is unlikely to be converted to adenosine, unlike ADP), which implies that it does not readily penetrate the blood–brain barrier. It is noteworthy that these ADPβS-induced vasodepressor responses were highly reproducible in all four groups, as they remained unchanged (*p* > 0.05) when the D–R curves were repeated two more times (Figure 2). This result validates our experimental conditions for pharmacological studies using up to two doses of a given antagonist.

### 3.5. Possible Participation of P2Y_1_, P2Y_12_, and P2Y_13_ Receptors in the Decreases in Diastolic Blood Pressure and the Increases in Systolic Blood Pressure Elicited by ADPβS

Based on the reproducibility of the diastolic vasodepressor (Figure 2) and systolic vasopressor (Figure 6) D–R curves for ADPβS, as well as the preferential agonist selectivity of ADPβS for P2Y_1_, P2Y_12_, and P2Y_13_ receptors [11,18,19,20,21], our next objective was to further investigate the pharmacological characteristics of the P2Y receptors involved in these responses. For this purpose, the P2Y receptor antagonists MRS2500 (P2Y_1_), PSB0739 (P2Y_12_), and MRS2211 (P2Y_13_) were used as pharmacological tools [9,10,11,18,27].

The doses selected for these antagonists in the present study: (i) have previously been justified in terms of their binding/affinity values and other properties by our group [21,30]; and (ii) have been demonstrated to cause a complete blockade of their respective receptor subtypes that elicit cardiovascular responses in pithed rats [21,30].

On this basis, the fact that 100 and 300 µg/kg of the P2Y_1_ receptor antagonist MRS2500 (with affinity values of **9.1**, 4.0, and 4.0 for, respectively, **P2Y_1_**, P2Y_12_, and P2Y_13_ receptors [18,21]) caused a complete blockade of the vasodepressor responses to ADPβS in all four groups (Figure 3) clearly suggests the role of P2Y_1_ receptor subtypes. This view is unequivocally strengthened by the fact that these ADPβS vasodepressor responses were resistant to blockade by: (i) the P2Y_12_ receptor antagonist PSB0739 (Figure 4), with affinity values of 6.0, **7.6,** and 6.0 for, respectively, P2Y_1_, **P2Y_12_**, and P2Y_13_ receptors [18,21]); or (ii) the P2Y_13_ receptor antagonist MRS2211 (Figure 5), with affinity values of 5.0, 5.0, and **6.3** for, respectively, P2Y_1_, P2Y_12_, and **P2Y_13_** receptors [18,21]). These P2Y_1_ receptors seem to show some similarities with other P2Y_1_ receptors reported to mediate endothelium-dependent vasodilatation via the nitric oxide–cGMP pathway in several isolated blood vessels [32,39,40,41].

Notwithstanding, the fact that the blocking profile produced by MRS2500 on these ADPβS diastolic vasodepressor responses was particularly different in anaesthetised rats with intact vagus nerves (Group 1) deserves further discussion. In this respect, 100 µg/kg MRS2500 (which practically abolished these responses in Groups 2, 3, and 4, in agreement with its high affinity value of 9.1 for P2Y_1_ receptors) significantly attenuated (but failed to abolish) these responses, while 300 µg/kg MRS2500 caused a complete blockade. This apparent discrepancy in Group 1 may be explained, at least in part, by the cardiovascular influence of the intact vagus nerves (unlike Group 2) and the activity of baroreflex compensatory mechanisms of central origin, which could have been triggered by the complex cardiovascular responses to ADPβS at two levels:

(1) At the vascular level, the decreases in DBP would trigger, in the CNS, baroreflex increases in sympathetic vascular (and cardiac) tone and decreases in vagal cardiac tone [1,3,4,5,6], which would tend to counteract these diastolic vasodepressor responses to ADPβS.

(2) At the cardiac (left ventricle) level, the increases in systolic blood pressure (Figure 1, Figure 6, and Figure 7) would trigger, in the CNS, baroreflex increases in vagal cardiac tone and decreases in sympathetic vascular and cardiac tones that decrease cardiac output [1,3,4,5,6]; this, in turn, would offset these ADPβS-induced systolic vasopressor responses (as discussed in Section 3.3 above). Indeed, these baroreflex compensatory mechanisms, including a decreased sympathetic vascular tone (which would produce systemic vasodilatation and, consequently, a decrease in DBP) may shed further light on why: (i) 300 µg/kg MRS2500 was required to abolish ADPβS-induced vasodepressor responses in Group 1 (i.e., with intact vagus nerves); and (ii) 100 µg/kg MRS2500 was enough to abolish these responses in Group 2 (i.e., with bilateral vagotomy).

Accordingly, keeping in mind the above lines of evidence and reasoning, we found it difficult to establish in anaesthetised animals (Groups 1 and 2): (i) the degree of contribution of the central baroreflex compensatory mechanisms triggered by the decreases in DBP and the increases in systolic blood pressure elicited by ADPβS; and (ii) how the potent vasodepressor responses to ADPβS predominated, despite the ultimate physiological balance of these central baroreflex compensatory mechanisms triggered by the peripheral systemic vasodilatation and cardiac contractility produced by ADPβS (clearly observed in Group 4).

On the other hand, regarding the pharmacological analysis of the P2Y receptors involved in ADPβS-induced increases in systolic blood pressure in Group 4, we considered it notable that these vasopressor responses were not dose-dependent and appeared to show a tachyphylactic (fading) pattern. Admittedly, this study did not investigate why systolic vasopressor responses would be more subject to tachyphylaxis than diastolic vasodepressor responses; notwithstanding, it is tempting to suggest a possible acute desensitization of cardiac (left ventricle) P2Y receptors. This suggestion is certainly in line with other studies showing desensitization of vascular P2Y_1_ receptors on endothelium involving mechanisms that depend on protein kinase C signalling [32]. In view of the fact that these ADPβS-induced increases in systolic blood pressure were reproducible, at least when repeating the D–R curve to ADPβS for a second time (Figure 6), these responses were pharmacologically analysed with the above antagonists, as previously described for the ADPβS-induced decreases in DBP. Interestingly, on the basis of the blocking potency observed with these antagonists (Figure 7), we can suggest: (i) the predominant role of P2Y_1_ receptors, because 100 µg/kg MRS2500 was enough to abolish these systolic vasopressor responses, with 300 µg/kg MRS2500 producing no further effects (*p* > 0.05); and (ii) a secondary role of P2Y_12_ and P2Y_13_ receptors, as PSB0739 (100 and 300 µg/kg) and MRS2211 (1000 and 3000 µg/kg) caused a dose-dependent blockade without abolishing the ADPβS responses (see above for antagonist affinities and other considerations regarding the blocking doses of antagonists). Moreover, our group has recently reported that ADPβS inhibits the rat vasodepressor CGRPergic tone by activation of P2Y_1_/P2Y_13_ receptors on perivascular sensory CGRPergic nerves [30]; accordingly, it remains to be investigated to what extent this inhibition of CGRP release in resistance blood vessels would affect the results of the present study.

Altogether, the two main findings of this study using the above P2Y receptor antagonists are:
(i)MRS2500, which binds with similar affinities at human and rat P2Y_1_ receptors [42], blocked both cardiovascular responses produced by ADPβS, namely, the decreases in DBP (i.e., systemic vasodilatation) in all four groups (Figure 3) and the increases in systolic blood pressure (i.e., cardiac left ventricular contractility) in Group 4 (Figure 7). This finding implies the role of P2Y_1_ receptors in both responses.(ii)PSB0739 and MRS2211 blocked the ADPβS-induced increases in systolic blood pressure (i.e., cardiac left ventricular contractility) in Group 4 (Figure 7), but not the ADPβS-induced decreases in DBP (which were blocked by MRS2500 in all four groups; Figure 3). This line of evidence suggests the additional secondary involvement of P2Y_12_ and P2Y_13_ receptors in ADPβS-induced cardiac left ventricular contractility. Within this context, since the differences in the affinity values of PSB0739 and MRS2211 for P2Y_1_, P2Y_12_, and P2Y_13_ receptors are around 1.5 log units (see above), one could argue that, at the i.v. doses used, the selectivity of these antagonists in pithed rats is rather low and, accordingly, that they also block (to a certain extent) P2Y_1_ receptors. If this were the case, however, PSB0739 and MRS2211 should have blocked the ADPβS-induced decreases in DBP (i.e., systemic vasodilatation mediated by MRS2500-sensitive P2Y_1_ receptors in all four groups), which was not the case (Figure 3).

### 3.6. Limitations of the Study

From a transductional and functional perspective, we recognise that the second messengers resulting from receptor activation, as well as the specific vascular location of the P2Y_1_ receptors mediating systemic vasodilatation with decreases in DBP in all four groups (Figure 1 and Figure 3) was not analysed in this investigation. These P2Y_1_ receptors are similar to other G*_q_*-coupled P2Y_1_ receptors that mediate endothelium-dependent vasodilatation in several isolated blood vessels via the nitric oxide–cGMP pathway activated by an endothelial increase in Ca^2+^ concentrations [9,10,11,20,27,32,39,40,41,43]. These limitations could be overcome by carrying out other studies that include the approaches of electrophysiology, biochemistry, immunohistochemistry and molecular biology techniques. A similar multidisciplinary approach could be applied for the transduction mechanisms/second messengers involved after activation of the P2Y receptors mediating the ADPβS-induced increases in systolic blood pressure (i.e., cardiac left ventricular contractility) in Group 4. In this respect, P2Y_1_ receptors are coupled to G*_q_*-proteins (associated with Ca^2+^ efflux from intracellular deposits) [9,10,11,27,43], and this mechanism may be consistent with cardiac contractility; in contrast, P2Y_12_ and P2Y_13_ receptors are coupled to G*_i/o_*-proteins [9,10,11,27,43], but other transduction mechanisms could be involved.

On the other hand, as previously considered [30], the affinities of ADPβS, MRS2500, PSB0739, and MRS2211 for P2Y_1_, P2Y_12_, and P2Y_13_ receptors (indicated above) were obtained from human P2Y receptors; nevertheless, these affinity data for rodents have not yet been published.

### 3.7. Perspectives and Potential Clinical Significance

As pointed out earlier [30], “P2Y receptors play an important role in numerous cardiovascular diseases including endothelial dysfunction, which is characterized by vasoconstriction, increased vascular permeability as well as a prothrombotic and proinflammatory state [20,40]”. Moreover, the fact that the P2Y_1_ receptor antagonist MRS2500 blocked the diastolic vasodepressor responses (i.e., systemic vasodilatation) and the systolic vasopressor responses (i.e., cardiac left ventricular contractility) induced by ADPβS [this study] clearly suggests that P2Y_1_ receptors are involved in cardiovascular regulation. In keeping with this suggestion, other studies have recently revealed that endothelial P2Y_1_ receptors are critical in mediating nitric oxide-dependent vasorelaxation [44] and in processes essential for the regeneration of damaged endothelium [45]. This knowledge has stimulated increased interest in considering the P2Y_1_ receptor as a potential therapeutic target in the treatment of hypertension [46].

Additionally, the field of purinergic antagonists is constantly evolving [47] and MRS2500 has only been used in one in vivo study [48], highlighting the need for further research on this notable specific compound, with clinical potential in both hypertension and thrombosis. Indeed, while activation of platelet P2Y**_1_** and P2Y**_12_** receptors may cause thrombosis [49], regulatory approval has been granted exclusively to P2Y**_12_** receptor antagonists. More recently, P2Y**_1_** receptor antagonists have been suggested for potential clinical use as anti-thrombotic agents [50]. Considering the outcomes of our study, it is evident that the vasodepressor responses to ADP involve activation of P2Y**_1_**, but not P2Y**_12_**, receptors [51]. This observation implies that the vasomotor impact of P2Y**_12_** receptor antagonists should be expected to be less pronounced than that of P2Y**_1_** receptor antagonists. Consequently, these findings hold significance in the decision-making process concerning the potential advancement of P2Y**_1_** receptor antagonists in thrombosis treatment.

## 4. Materials and Methods

### 4.1. Ethical Endorsement of the Experimental Protocols in Anaesthetised and Pithed Rats

As reported elsewhere [21,30,35], the experimental protocols of the present investigation were endorsed by our Institutional Ethics Committee on the use of animals on scientific experiments (CICUAL-Cinvestav; protocol number 0139-15), following the regulations established by the Mexican Official Norm [NOM-062-ZOO-1999] [52] in accordance with the guide for the Care and Use of Laboratory Animals in the USA [53], the ARRIVE guidelines for reporting experiments in animals [54], and the Directive 2010/63/EU of the European parliament on the protection of animals used for scientific purposes.

### 4.2. General Methods

A total of 120 male normotensive Wistar rats was used in the present investigation. As shown in Figure 8, these animals encompassed two main experimental sets to produce vasodepressor responses, namely: (i) anaesthetised rat model, consisting of 60 animals with a body weight between 300–350 g, as described by De Vries et al. [36]; and (ii) pithed rat model, comprising 60 animals with a body weight between 380–420 g, as recently reported [30]. “The animals were housed under controlled environmental conditions, with a room temperature of 22 ± 2 °C, a humidity level of 50%, and a light/dark cycle of 12 h of light followed by 12 h of darkness. The light phase of the cycle began at 07:00 h. Throughout the study, the animals had unrestricted access to food and water, which were provided in their respective home cages”.

On the day of the experiments, all rats were initially anaesthetised by means of an intraperitoneal injection of sodium pentobarbital (60 mg/kg); the suitability of this anaesthetic regimen prior to any surgical procedures has been validated in other studies in rats [30,37]. Then, the 120 rats were cannulated with a cannula into the trachea for artificial respiration [30,36], and underwent the following experimental procedures.

***(i) Anaesthetised rat model (n = 60).*** As reported earlier [35,36], “following successful cannulation of the trachea, the animals were then connected to an Ugo Basile pump (Ugo Basile Srl, Comerio, VA, Italy) to provide artificial breathing using room air. The ventilation parameters were set at a rate of 56 strokes/min, with a stroke volume of 20 mL/kg of body weight. Then, polyethylene catheters were placed in: (i) the left femoral vein for administration of the vehicle (bidistilled water, 1 mL/kg) or ADPβS (0.3–10 µg/kg); (b) the right femoral vein for administration of the purinergic P2Y receptor antagonists; (iii) the left carotid artery to connect it to a Grass pressure transducer (P23 XL), which was designed for the purpose of measuring and documenting blood pressure. Simultaneous recordings of heart rate (measured with a 7P4F tachograph) and blood pressure were obtained using a model 7D Grass polygraph manufactured by Grass Instrument Co. (Quincy, MA, USA)”. Subsequently, this set was randomly divided into two groups (*n* = 30 each; see Figure 8), i.e.,: (i) Group 1 (anaesthetised rats without vagotomy), which remained with both vagus nerves intact to analyse the possible influence of the CNS on the blood pressure effects of ADPβS via vagal compensatory reflex mechanisms; and (ii) Group 2 (anaesthetised rats with vagotomy), which underwent a bilateral cervical vagotomy to avoid the possible central influence of reflex responses resulting from activation of the vagus nerve (e.g., the von Bezold–Jarisch-like reflex responses [55]) in response to the blood pressure effects of ADPβS. It is noteworthy that additional slow i.v. injections of 10–20 mg/kg pentobarbital were given every hour (typically after each D–R curve to ADPβS) to maintain the level of anaesthesia in this set (*n* = 60).

***(ii) Pithed rat model (n = 60).*** As previously described [30,37,38], “after cannulation of the trachea the rats underwent the process of pithing, which requires the insertion of a stainless steel rod through the orbit and foramen magnum into the vertebral foramen. Subsequently, the animals were, in sequence: (i) artificially ventilated with room air by an Ugo Basile pump as described above; and (ii) bilaterally vagotomised at cervical level. Then, as pointed out earlier for the anaesthetised rat model, catheters were placed in the left and right femoral veins (for administration of compounds), as well as in the left carotid artery (to record blood pressure)”. Afterwards, the animals were divided into two groups (*n* = 30 each; see Figure 8), namely: (i) Group 3 (pithed rats); and (ii) Group 4 (pithed rats continuously infused [i.v.] with methoxamine). Only in Group 4 was “20 µg/kg.min methoxamine infused through a cannula inserted into the left femoral vein to increase and maintain DBP at an approximate value of 115 mm Hg during and until the end of the experiments”; this procedure restores the systemic vascular tone to levels similar to those observed in anaesthetised rats (see above), as previously described [30,37,38].

A lamp kept the body temperature of all rodents (measured with rectal thermometers) at 37 °C.

### 4.3. Experimental Protocols

Once the rats were haemodynamically stable for 30 min, DBP and heart rate baseline values were assessed in the four groups. Subsequently, each group was subdivided into five different treatment subgroups (*n* = 6 for each; see Figure 8), as outlined below.

#### 4.3.1. Protocol I: The Effects of Consecutive i.v. Bolus of Vehicle or ADPβS on Blood Pressure

As illustrated in Figure 8, 12 rats of Group 1, Group 2, Group 3, and Group 4 (*n* = 48 in total) were separated into two subgroups, each consisting of six rats, in order to investigate the effects on blood pressure and heart rate after administration of consecutive i.v. bolus of: (i) vehicle (bidistilled water, 1 mL/kg administered seven times); and (ii) ADPβS (0.3, 0.56, 1, 1.8, 3, 5.6, and 10 µg/kg). These D–R curves to the vehicle and ADPβS were repeated three times to analyse the reproducibility of these effects and the stability of our experimental haemodynamic conditions. In Group 4, the D–R curves were started after DBP was at a stable approximate value of 115 mm Hg during the methoxamine infusion, as mentioned above.

#### 4.3.2. Protocol II: Effect of the Antagonists MRS2500 (P2Y1), PSB0739 (P2Y12), or MRS2211 (P2Y13) on the Blood Pressure Changes Caused by i.v. Bolus of ADPβS

Eighteen animals of Group 1, Group 2, Group 3, and Group 4 (*n* = 72 in total) were subdivided into three subgroups (*n* = 6 for each) to evaluate the blood pressure responses produced by an i.v. bolus of ADPβS (0.3, 0.56, 1, 1.8, 3, 5.6, and 10 µg/kg) before and after i.v. administration of: (i) MRS2500 (100 and 300 µg/kg); (ii) PSB0739 (100 and 300 µg/kg); and (iii) MRS2211 (1000 and 3000 µg/kg) (see Figure 8). It has previously been shown that these doses of antagonists are sufficiently high to completely block their respective receptors that mediate cardiovascular responses in rats [21,30].

### 4.4. Compounds

The following substances were used in the current study (obtained from the sources listed), as previously reported [17,21,30]: “sodium pentobarbital (PISA Agropecuaria, Mexico City, Mexico); methoxamine hydrochloride and adenosine-5′-[β-thio]diphosphate trilithium salt (ADPβS) (Sigma Chemical Co., St. Louis, MO, USA); (1R*,2S*)-4-[2-Iodo-6-(methylamino)-9H-purin-9-yl]-2-(phosphonooxy) bicyclo [3.1.0]hexane-1-methanol dihydrogen phosphate ester tetra ammonium salt (MRS2500); 2-[(2-chloro-5-nitrophenyl)azo]-5-hydroxy-6-methyl-3-[(phosphonooxy)methyl]–4-pyridinecarboxaldehyde disodium salt (MRS2211) (TOCRIS, Avonmouth, Bristol, UK) and 1-amino-9,10-dihydro-9,10-dioxo-4-[[4-(phenylamino)-3-sulfophenyl]amino]-2-anthracenesulfonic acid sodium salt (PSB0739). ADPβS, MRS2500, PSB0739, and MRS2211 were dissolved in bidistilled water, as indicated in other studies [17,21,30]”.

### 4.5. Presentation of Data and Statistical Analysis

The results are presented as mean ± S.E.M. The peak changes in blood pressure produced by ADPβS consisted of: (i) immediate diastolic vasodepressor responses in all four groups, which were expressed as the percentage decreases in diastolic blood pressure relative to baseline readings [37]; and (ii) subsequent systolic vasopressor responses in Group 4, which were calculated as the absolute increases in systolic blood pressure relative to baseline readings [37]. These values were used to quantify the effects of the different treatments. The difference between the changes in diastolic (or systolic) blood pressure within a subgroup of animals was evaluated with the Student–Newman–Keuls’ test when a two-way repeated measures ANOVA (randomised block design) indicated that the samples were from different populations [56]. The threshold for statistical significance was deemed to be met when the *p*-value was less than 0.05 (two-tailed). The statistical analysis was conducted using SigmaPlot 12.0, a software developed by Systat Software, Inc. specifically designed for Windows operating systems.

The graphs were generated with Prism 6 software, developed by GraphPad Software, Inc. (San Diego, CA, USA).

## 5. Conclusions

Our findings, altogether, demonstrate in male Wistar rats that ADPβS can decrease diastolic blood pressure in the absence and presence of a functional CNS (i.e., despite triggering central baroreflex compensatory mechanisms) by exclusive peripheral activation of purinergic P2Y_1_ receptors. In contrast, in the absence of a functional CNS and restoration of the systemic vascular tone with methoxamine, ADPβS may increase systolic blood pressure by peripheral activation of purinergic P2Y_1_, P2Y_12_, and P2Y_13_ receptors.

## Figures and Tables

**Figure 1 pharmaceuticals-16-01683-f001:**
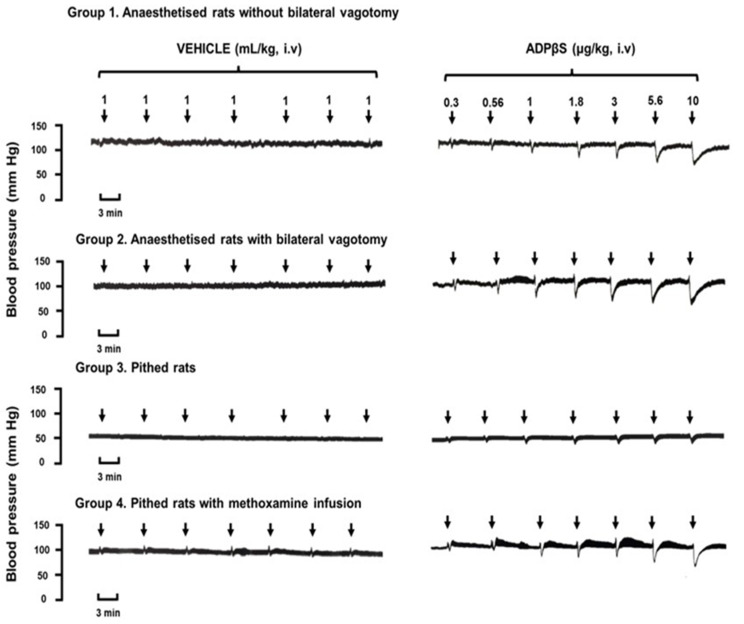
Prototypical experimental recordings of the responses to either the vehicle (**left** panel) or ADPβS (**right** panel). Responses produced by the i.v. bolus of the vehicle (bidistilled water; 1 mL/kg given seven times) or ADPβS (0.3, 0.56, 1, 1.8, 3, 5.6, and 10 µg/kg) on blood pressure in male Wistar rats under different experimental conditions, namely: (i) anaesthetised rats without bilateral vagotomy (Group 1); (ii) anaesthetised rats with bilateral vagotomy (Group 2); (iii) pithed rats (Group 3); and (iv) pithed rats continuously infused (i.v.) with 20 µg/kg·min of methoxamine (Group 4).

**Figure 2 pharmaceuticals-16-01683-f002:**
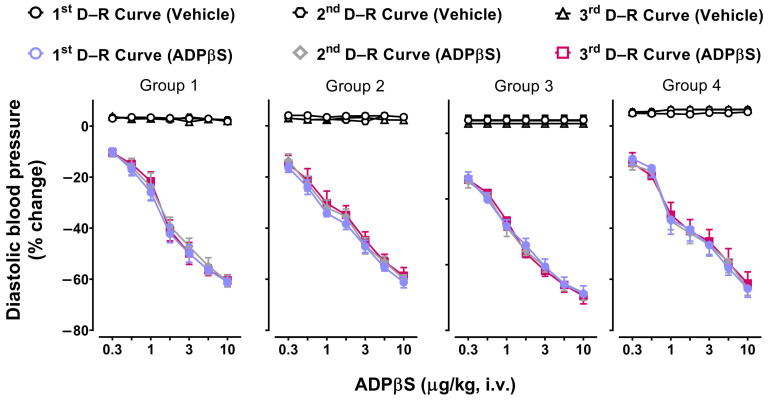
Reproducibility of the responses induced by the vehicle or ADPβS on diastolic blood pressure (percent change). Effects produced by consecutive i.v. bolus of either the vehicle (bidistilled water, 1 mL/kg given seven times; black empty symbols) or ADPβS (0.3, 0.56, 1, 1.8, 3, 5.6, and 10 µg/kg; coloured symbols) on diastolic blood pressure (D–R curves were repeated three times for each compound) in anaesthetised rats without bilateral vagotomy (Group 1), anaesthetised rats with bilateral vagotomy (Group 2), pithed rats (Group 3), and pithed rats continuously infused (i.v.) with methoxamine (Group 4) (*n =* 6 for each group). Unfilled symbols represent vehicle responses (◯, **⎔**, **△**), whereas filled symbols (●, ◆, ■) denote a significant difference (*p <* 0.05) in ADPβS responses compared with the corresponding vehicle responses. There were no significant differences (*p* > 0.05) amongst the three D–R curves produced by either the vehicle or ADPβS. Values represent means ± SEM (*n* = 6 for each group).

**Figure 3 pharmaceuticals-16-01683-f003:**
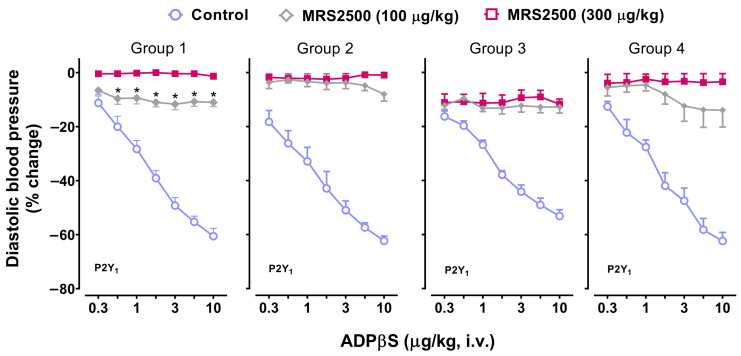
Effect of the P2Y_1_ receptor antagonist on the decreases in diastolic blood pressure (% change) elicited by ADPβS Effect of i.v. administration of MRS2500 (100 and 300 μg/kg) on the vasodepressor responses (determined as peak changes) elicited by ADPβS (0.3, 0.56, 1, 1.8, 3, 5.6, and 10 µg/kg, i.v.) in anaesthetised rats without bilateral vagotomy (Group 1), anaesthetised rats with bilateral vagotomy (Group 2), pithed rats (Group 3), and pithed rats continuously infused (i.v.) with methoxamine (Group 4). Unfilled symbols denote either control responses (O) or a nonsignificant response (◇) compared to the corresponding control response (*p >* 0.05). Filled symbols (◆, ■) exemplify significant differences (*p* < 0.05) against the corresponding control responses (O). (*) Indicates significant differences (*p <* 0.05) between the vasodepressor responses produced by 100 and 300 μg/kg MRS2500. Values represent means ± SEM (*n* = 6 for each group).

**Figure 4 pharmaceuticals-16-01683-f004:**
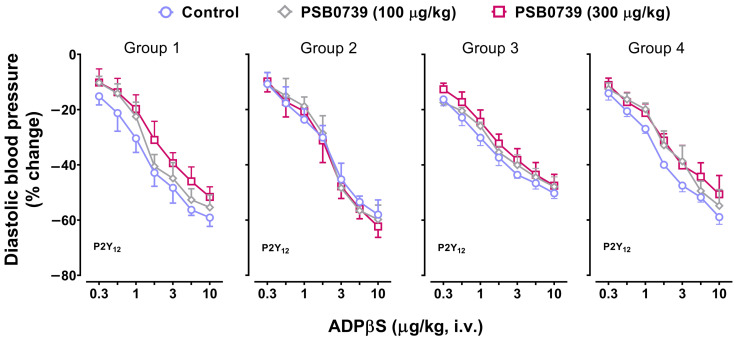
Effect of the P2Y_12_ receptor antagonist on the decreases in diastolic blood pressure (% change) elicited by ADPβS. The effect of i.v. administration of PSB0739 (100 and 300 μg/kg) on the vasodepressor responses (determined as peak changes) elicited by ADPβS (0.3, 0.56, 1, 1.8, 3, 5.6, and 10 µg/kg, i.v.) in anaesthetised rats without bilateral vagotomy (Group 1), anaesthetised rats with bilateral vagotomy (Group 2), pithed rats (Group 3), and pithed rats continuously infused (i.v.) with methoxamine (Group 4). Unfilled symbols denote either control responses (O) or nonsignificant responses (◇, ☐) compared to the corresponding control responses (*p >* 0.05). Values represent means ± SEM (*n* = 6 for each group).

**Figure 5 pharmaceuticals-16-01683-f005:**
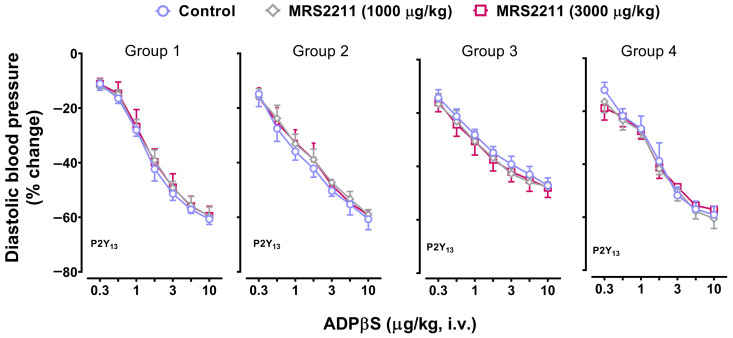
Effect of the P2Y_13_ receptor antagonist on the decreases in diastolic blood pressure (% change) elicited by ADPβS. Effect of i.v. administration of MRS2211 (1000 and 3000 μg/kg) on the vasodepressor responses (determined as peak changes) elicited by ADPβS (0.3, 0.56, 1, 1.8, 3, 5.6, and 10 µg/kg, i.v.) in anaesthetised rats without bilateral vagotomy (Group 1), anaesthetised rats with bilateral vagotomy (Group 2), pithed rats (Group 3), and pithed rats continuously infused (i.v.) with methoxamine (Group 4) (*n =* 6 for each group). Unfilled symbols denote either control responses (O) or nonsignificant responses (◇, ☐)) compared to the corresponding control responses (*p >* 0.05). Values represent means ± SEM (*n* = 6 for each group).

**Figure 6 pharmaceuticals-16-01683-f006:**
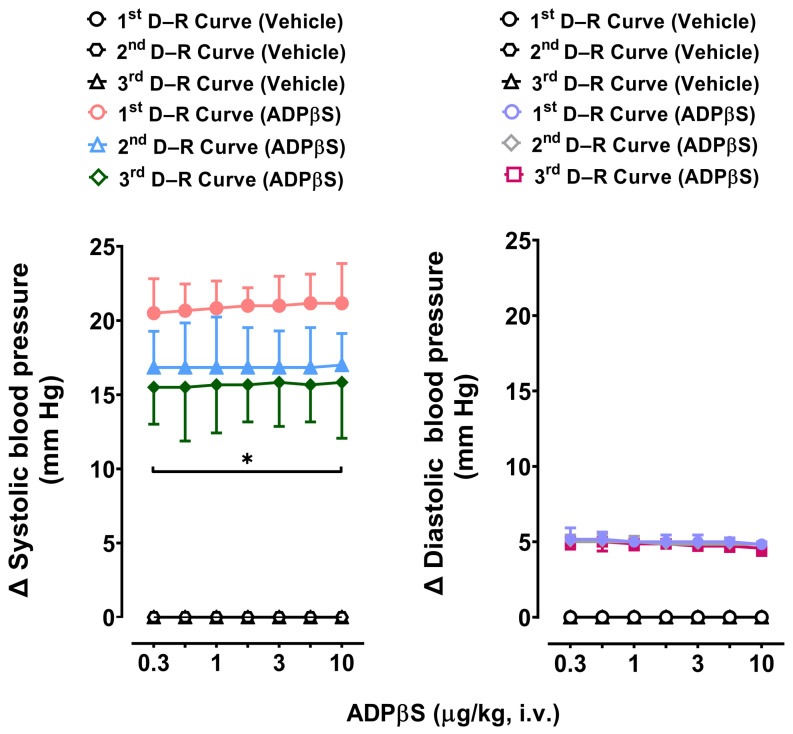
The systolic (**left** panel) and diastolic (**right** panel) blood pressure changes in pithed rats continuously infused (i.v.) with methoxamine (Group 4). The effects elicited by an i.v. bolus of the vehicle (1 mL/kg given seven times; *n* = 6) or ADPβS (0.3, 0.56, 1, 1.8, 3.1, 5.6, and 10 µg/kg; *n* = 6) on systolic and diastolic blood pressure in pithed rats continuously infused (i.v.) with 20 μg/kg.min of methoxamine (Group 4). Filled symbols denote a significant difference (*p <* 0.05) compared to the corresponding i.v. bolus of the vehicle (1 mL/kg). (*) The third D–R curve of the vasopressor responses to ADPβS (**left** panel) was significantly attenuated (*p* < 0.05) when contrasted with the first (control) D–R curve.

**Figure 7 pharmaceuticals-16-01683-f007:**
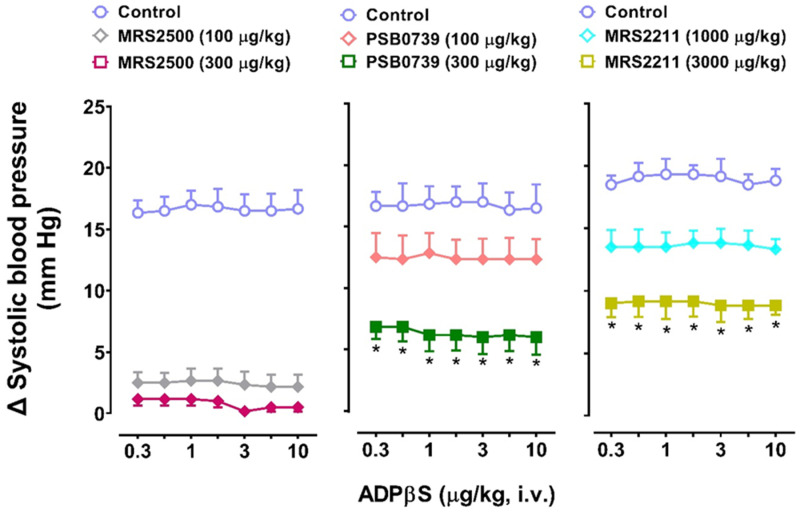
Effects of the antagonists MRS2500 (P2Y_1_), PSB0739 (P2Y12), or MRS2211 (P2Y_13_) on the systolic vasopressor responses elicited by ADPβS in Group 4. The effects produced by MRS2500 (100 and 300 µg/kg; left panel), PSB0739 (100 and 300 µg/kg; middle panel) or MRS2211 (1000 and 3000 μg/kg; right panel) given i.v. on the increases in systolic blood pressure elicited by an i.v. bolus of ADPβS (0.3, 0.56, 1, 1.8, 3, 5.6, and 10 µg/kg) in pithed rats continuously infused (i.v.) with methoxamine (Group 4). Unfilled symbols represent control responses (O). Filled symbols (◆, ■) denote significant differences (*p* < 0.05) compared to the corresponding control responses (O). (*) Indicates significant differences (*p <* 0.05) in the vasopressor responses to ADPβS between i.v. administration of 100 and 300 μg/kg PSB0739 (middle panel) or 1000 and 3000 μg/kg MRS2211 (right panel). Values represent means ± SEM (*n* = 6 for each group).

**Figure 8 pharmaceuticals-16-01683-f008:**
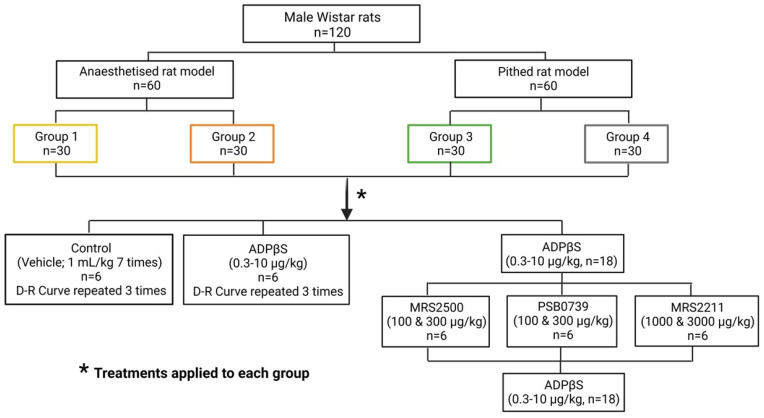
Experimental Design. Distribution of the animals (*n* = 120) in the different protocols indicating the specific number of animals utilised in the two experimental models (i.e., anaesthetised and pithed; *n* = 60 each) and, subsequently, their division into: Group 1 (anaesthetised rats without vagotomy; Group 2 (anaesthetised rats with bilateral vagotomy); Group 3 (pithed rats); and Group 4 (pithed rats continuously infused [i.v.] with 20 µg/kg.min of methoxamine. Note that each group (*n* = 30) was further subdivided into five subgroups (*n* = 6 for each) that underwent the same pharmacological treatments.

**Table 1 pharmaceuticals-16-01683-t001:** Diastolic blood pressure (DBP) values in the four groups of rats 10 min after i.v. treatment with bidistilled water (vehicle to dissolve all compounds) or antagonists. The control group received no treatment.

Treatment	Doses	DBP (mm Hg)			
		Anaesthetised Rats without Vagotomy (Group 1)	Anaesthetised Rats with Bilateral Vagotomy (Group 2)	Pithed Rats (Group 3)	Pithed Rats Continuously Infused (i.v.) with Methoxamine (Group 4)
Control	No treatment ^a^	104 ± 3	103 ± 5	37 ± 4	113 ± 4
Vehicle	1 mL/kg ^b^	105 ± 2	103 ± 5	38 ± 3	113 ± 2
ADPβS	0.3–10 µg/kg ^c^	105 ± 5	103 ± 2	33 ± 2	115 ± 4
MRS2500	300 µg/kg	111 ± 5	120 ± 5	46 ± 4	112 ± 4
PSB0739	100 µg/kg	114 ± 5	114 ± 4	45 ± 4	119 ± 5
PSB0739	300 µg/kg	102 ± 4	116 ± 3	43 ± 5	121 ± 2
MRS2211	1000 µg/kg	102 ± 4	112 ± 4	45 ± 5	120 ± 6
MRS2211	3000 µg/kg	104 ± 6	112 ± 4	44 ± 4	117 ± 4

^a^ Corresponds to DBP values before the first D–R curve to the vehicle (bidistilled water) in Subgroup 1 (see Section 4.3 of methods). ^b^ Corresponds to DBP values after the first i.v. bolus of the vehicle in the first D–R curve for the vehicle in Subgroup 1 (see Section 4.3 of methods); the six subsequent i.v. boluses of the vehicle also had no significant effects. ^c^ Corresponds to DBP values after the first D–R curve for ADPβS (total cumulative dose: 22.26 µg/kg) in Subgroup 2 (see Section 4.3 of methods); the two subsequent D–R curves for ADPβS also had no significant effects.

## Data Availability

Data is contained within the article.

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
