# Peer review of "Pharmacological Nature of the Purinergic P2Y Receptor Subtypes That Participate in the Blood Pressure Changes Produced by ADPβS in Rats"

_pharmaceuticals, 2023, doi:10.3390/ph16121683_

Round 1

Reviewer 1 Report

Comments and Suggestions for Authors

The authors present the results of a carefully designed study and how they investigated the hypothesis of their study. Analysis of the research was as well carefully planned and done. Tables and figures present all relevant results of the research. The discussion part puts the findings of the authors in the appropriate context and describes how the authors did get to their conclusions. The results are of importance to stimulate future research.

Author Response

Thank you very much for your quick and encouraging reviewer report. With all best wishes,

Carlos M. Villalón

Reviewer 2 Report

Comments and Suggestions for Authors

The paper explores the role of purinergic P2Y receptor subtypes in ADPβS-induced blood pressure changes in rats, both centrally and peripherally. Several P2Y receptor antagonists were used to assess their pharmacological properties. The findings align closely with prior research and hold potential clinical significance. Nevertheless, there are notable shortcomings that warrant attention.

1.     Abstract: The abstract offers a concise summary of the paper's objectives. However, the opening sentence can be simplified for better clarity, and the findings could be more summarily presented.

2.     Introduction: The introduction establishes a clear context for the study, emphasizing the importance of comprehending purinergic P2Y receptor subtypes and their role in blood pressure regulation. While it could benefit from greater context through references to previous research. Additionally, it would be valuable to clarify the distinctions between this study and the previous one (reference 25). Is this a continuous exploration of different scientific questions?

3.     Results: The presentation of results is commendable, with clear and well-organized visuals.

4.     Discussion: The discussion thoughtfully explores the results and the contribution of purinergic receptors to blood pressure regulation, while also acknowledging potential limitations. However, it could be enriched by delving into recent previous studies to underscore the significance of this study's findings.

5.     Methods: The methodology is meticulously described and highly readable.

6.     Citations and References: Citations and references are properly formatted and up-to-date.

In conclusion:

The paper investigates the role of P2Y receptor subtypes in ADPβS-induced blood pressure changes in rats. It does so by monitoring systolic and diastolic blood pressure changes in rats treated with P2Y receptor subtype antagonists. However, the study lacks histological or more direct evidence, which limits the robustness of its findings. Additional research in this field is strongly recommended.

Comments on the Quality of English Language

Some English expressions are complex and may benefit from improvement by a native speaker.

Author Response

  1. Abstract: The abstract offers a concise summary of the paper's objectives. However, the opening sentence can be simplified for better clarity, and the findings could be more summarily presented.

Our reply: As suggested, the opening sentence has been simplified (see Abstract: page 1, lines 23-24). Moreover, some sentences have been made clearer. Since the total number of words in our abstract now consists of 194 words, we would prefer to keep the abstract as shown for the sake of clarity.

  1. Introduction: The introduction establishes a clear context for the study, emphasizing the importance of comprehending purinergic P2Y receptor subtypes and their role in blood pressure regulation. While it could benefit from greater context through references to previous research. Additionally, it would be valuable to clarify the distinctions between this study and the previous one (reference 25). Is this a continuous exploration of different scientific questions?

Our reply: Suggestion taken. Hence, we have inserted the following paragraphs in the introduction to further widen its context with 5 new references (see page 2, lines 96-99; and page 3, lines 100-106):

“In this context, some lines of evidence suggest that the purinergic P2Y1 receptor plays a role in blood pressure regulation, namely: (i) blockade of this receptor in neurons leads to decreased peripheral chemoreceptor-mediated activation, impacting blood pressure control [22]; and (ii) this receptor participates in urinary NaCl excretion under high-sodium diets [23] and decreases pulmonary arterial pressure in pulmonary hypertension cases [24]. However, the direct effects resulting from activation of peripheral P2Y1 receptors have not been investigated in detail.

Remarkably: (i) P2Y1 receptor deficiency in diabetic rat models, particularly in mesenteric arteries, suggests its potential role in diabetes-related vascular complications [25]; and (ii) P2Y1 receptor knockout mice show reduced atherosclerosis markers, emphasizing its importance in vascular health [26]”.

On the other hand, the present study is completely different from the one quoted as reference 25 (now reference 30 after the inclusion of additional references). This last paper analyses the pharmacological profile of the P2Y receptors that inhibit (in response to ADPβS) the vasodepressor sensory CGRPergic outflow, i.e. the stimulation by ADPβS of the prejunctional P2Y receptors on perivascular CGRPergic sensory nerves that innervate the systemic vasculature, which results in inhibition of CGRP release. Accordingly, we would prefer to keep the introduction without further changes to avoid confusion.

  1. Results: The presentation of results is commendable, with clear and well-organized visuals.

Our reply: We very much appreciate your comment.

  1. Discussion: The discussion thoughtfully explores the results and the contribution of purinergic receptors to blood pressure regulation, while also acknowledging potential limitations. However, it could be enriched by delving into recent previous studies to underscore the significance of this study's findings.

Our reply: As suggested, we have inserted the following paragraphs quoting 5 recent studies to broaden our discussion context (see page 17, lines 580-589):

“In keeping with this suggestion, other studies have recently revealed that endothelial P2Y1 receptors are critical in mediating nitric oxide-dependent vasorelaxation [44] and in processes essential for the regeneration of damaged endothelium [45]. This knowledge has stimulated increased interest in considering the P2Y1 receptor as a potential therapeutic target in the treatment of hypertension [46].

Additionally, the field of purinergic antagonists is constantly evolving [47] and MRS2500 has only been used in one in vivo study [48], highlighting the need for further research on this notable specific compound, with clinical potential in both hypertension and thrombosis”.

  1. Methods: The methodology is meticulously described and highly readable.

Our reply: We very much appreciate your comment.

  1. Citations and References: Citations and references are properly formatted and up-to-date.

Our reply: We very much appreciate your comment.

In conclusion:

The paper investigates the role of P2Y receptor subtypes in ADPβS-induced blood pressure changes in rats. It does so by monitoring systolic and diastolic blood pressure changes in rats treated with P2Y receptor subtype antagonists. However, the study lacks histological or more direct evidence, which limits the robustness of its findings. Additional research in this field is strongly recommended.

Our reply: In the context of the introduction and objectives of our study, this reviewer conclusion would seem to underestimate our blood pressure measurements and overvalue histological (and other) evidence. Although mesenteric vascular tone is a determining factor of peripheral vascular resistance, we would like to highlight that blood pressure is directly proportional to cardiac output and peripheral vascular resistance; and the last variable is, in turn, the algebraic sum of vascular tones in each and every vascular bed in the body. Consequently, what happens in a specific blood vessel within a vascular bed (with histological evidence) will not necessarily be reflected in blood pressure changes. Accordingly, it would be desirable to carry out other studies that include the approach of electrophysiology, biochemistry, immunohisto-chemistry and molecular biology techniques to complement our study. Certainly, these limitations of our study had already been clearly recognized in the original manuscript and now appear in section 3.6. Limitations of the study (see page 16, lines 550-566).    

Comments on the Quality of English Language: Some English expressions are complex and may benefit from improvement by a native speaker.

Our reply: Our manuscript has been revised for English grammar and style by a colleague fluent in English writing.

Reviewer 3 Report

Comments and Suggestions for Authors

In the original article entitled: „Pharmacological nature of the purinergic P2Y receptor subtypes 2 that participate in the blood pressure changes produced by ADPβS in the rat” by dr Silva-Velasco et al., the Authors investigated the involvement of various subtypes of P2Y receptors in mediating the blood pressure alterations induced by ADPβS administered intravenously in the rat model. In my opinion, the study is properly designed which is nicely illustrated in the Figure 8. Furthermore, the paper is extremely well-written. I would like to congratulate the Authors a very good work. I have only a couple of minor comments listed below.

1. The Authors used parametric tests, such as Student-Newman-Keuls test as post-hoc test after ANOVA, however the conditions for these test are a normality of data distribution and homogeneity of variances. I would like to ask the Authors whether they checked these conditions, if so, it should be added (with the names of respective tests) to the description of statistical analysis.

2. There is no need to introduce i.v. as the abbreviation of “intravenous” in the abstract. Please just leave the full name.

Author Response

In the original article entitled: „Pharmacological nature of the purinergic P2Y receptor subtypes 2 that participate in the blood pressure changes produced by ADPβS in the rat” by dr Silva-Velasco et al., the Authors investigated the involvement of various subtypes of P2Y receptors in mediating the blood pressure alterations induced by ADPβS administered intravenously in the rat model. In my opinion, the study is properly designed which is nicely illustrated in the Figure 8. Furthermore, the paper is extremely well-written. I would like to congratulate the Authors a very good work. I have only a couple of minor comments listed below.

Our reply: Thank you very much for your encouraging reviewer report.

  1. The Authors used parametric tests, such as Student-Newman-Keuls test as post-hoc test after ANOVA, however the conditions for these test are a normality of data distribution and homogeneity of variances. I would like to ask the Authors whether they checked these conditions, if so, it should be added (with the names of respective tests) to the description of statistical analysis.

Our reply: We have now edited and completed this section as follows (see page 20, lines 729-742):

Results are presented as mean ± S.E.M. The peak changes in blood pressure produced by ADPβS consisted of: (i) immediate diastolic vasodepressor responses in all four groups, which were expressed as the percentage decreases in diastolic blood pressure relative to baseline readings [37]; and (ii) subsequent systolic vasopressor responses in group 4, which were calculated as the absolute increases in systolic blood pressure relative to baseline readings [37]. These values were used to quantify the effects of the different treatments. The difference between the changes in diastolic (or systolic) blood pressure within a subgroup of animals was evaluated with the Student-Newman-Keuls’ test when a two-way repeated measures ANOVA (randomized block design) indicated that the samples were from different populations [56]. The threshold for statistical significance was deemed to be met when the p-value was less than 0.05 (two-tailed).

  1. There is no need to introduce i.v. as the abbreviation of “intravenous” in the abstract. Please just leave the full name.

Our reply: As suggested, we have omitted the abbreviation “i.v.” in the abstract.
